# Vascular Niche Facilitates Acquired Drug Resistance to c-Met Inhibitor in Originally Sensitive Osteosarcoma Cells

**DOI:** 10.3390/cancers14246201

**Published:** 2022-12-15

**Authors:** Weifeng Tang, Yu Zhang, Haixia Zhang, Yan Zhang

**Affiliations:** MOE Key Laboratory of Gene Function and Regulation, State Key Laboratory of Biocontrol, School of Life Sciences, Sun Yat-sen University, Guangzhou 510006, China

**Keywords:** osteosarcoma, c-Met, drug resistance, vascular niche, tumor heterogeneity

## Abstract

**Simple Summary:**

Osteosarcoma (OS) is a common malignant bone tumor in adolescents whose survival rates have not improved over the past few decades. At present, there is a lack of effective molecule-targeted therapy. Abnormal activation of c-Met is often detected in patients with OS, and c-Met inhibitors are considered to have tumor suppressive potential. However, OS is prone to acquire resistance to c-Met inhibitor, and the mechanisms of drug resistance are poorly understood. The goal of this study is to explore the mechanism of resistance to c-Met-targeted inhibitors. Our results demonstrate that the neovascular microenvironment contributes to the resistance of OS to c-Met-targeted therapy. Simultaneous targeting of c-Met and VEGFR2 can effectively overcome drug resistance.

**Abstract:**

Osteosarcoma (OS) is the most common primary bone tumor in children and adolescents characterized by drug resistance and poor prognosis. As one of the key oncogenes, c-Met is recognized as a promising therapeutic target for OS. In this report, we show that c-Met inhibitor PF02341066 specifically killed OS cells with highly phosphorylated c-Met in vitro. However, the inhibitory effect of PF02341066 was abrogated in vivo due to interference from the vascular niche. OS cells adjacent to microvessels or forming vascular mimicry suppressed c-Met expression and phosphorylation. Moreover, VEGFR2 was activated in OS cells and associated with acquired drug resistance. Dual targeting of c-Met and VEGFR2 could effectively shrink the tumor size in a xenograft model. c-Met-targeted therapy combined with VEGFR2 inhibition might be beneficial to achieve an ideal therapeutic effect in OS patients. Together, our results confirm the pivotal role of tumor heterogeneity and the microenvironment in drug response and reveal the molecular mechanism underlying acquired drug resistance to c-Met-targeted therapy.

## 1. Introduction

Tumor heterogeneity has been recognized as a significant challenge in the development of personalized cancer medicine. Phenotypic and functional variations among different cancer types as well as cancer cells within individual tumors arise in multiple ways. Accumulating cellular genetic and epigenetic changes during the tumorigenic process give rise to tumor heterogeneity, which is also influenced by microenvironmental signals during tumor progression [1]. Microenvironmental signals that consist of different cell types and soluble factors have been proven to foster diverse cancer cell properties within the same tumor [2,3]. Importantly, the well-established cancer stem cell (CSC) model is also thought to shape the intratumor heterogeneity that drives tumor progression and treatment failure [4].

Osteosarcoma (OS) is the most common malignant primary bone tumor in children and adolescents and is characterized by abundant angiogenesis and pulmonary metastasis [5]. Nearly one third of OS patients experience recurrence or progression and have limited treatment options and poor outcomes [6]. Compared to normal bone tissue, OS presents abnormal vascular mimicry (VM) and an increase in microvessels. Microvessel density (MVD) is significantly and positively related to a higher grade of malignancy and poor patient outcomes [7,8]. OS comprises genotypically and phenotypically heterogeneous cancer cells with considerable diversity in histologic features and malignant grade [9]. Communication and interaction with the bone microenvironment underlie OS initiation and progression while engendering tumor heterogeneity. A previous study demonstrated that abundant transforming growth factor β1 (TGFβ1) and a hypoxic environment could induce adherent OS cells toward a CSC phenotype, which was termed osteosarcoma stem cell (OSC). This CSC subpopulation was characterized by elevated tumorigenicity, neo-vasculogenesis, chemoresistance and metastatic potential [10,11]. The complex heterogeneity of OS has severely hampered clinical success and accounts for its poor prognosis. Few targeted drugs have thus far been utilized as effective clinical treatments for OS [12].

Among the attempts to develop new agents for the treatment of OS, molecular targeting of the c-Met receptor is regarded as a promising therapeutic strategy showing encouraging results. c-Met was originally identified as a protein product of the translocated promoter region (TPR)-MET transforming oncogene in the chemically transformed human OS cell line MNNG/HOS [13]. As a cell surface receptor tyrosine kinase (RTK) for hepatocyte growth factor/scatter factor (HGF/SF) [14], its overexpression is frequently detected in OS clinical specimens [15]. Overexpression of c-Met could induce primary human osteoblasts and mesenchymal cells to acquire malignant phenotypes [16,17]. The introduction of dominant-negative c-Met could significantly suppress OS proliferation, invasion and tumorigenicity [18]. Further studies indicated that inhibition of c-Met phosphorylation could downregulate the activation of Erk and Akt and suppress OS malignant behavior [19,20,21].

The small-molecule inhibitor crizotinib, also termed PF02341066, has been shown to be a potent selective ATP-competitive inhibitor of c-Met kinase across a panel of human tumors [22] and has been tested in phase II/III clinical trials of non-small-cell lung cancer [23]. Mechanisms of resistance to c-MET-targeted tyrosine kinase inhibitors have been partially analyzed and mainly attributed to the mutation, fusion, and focal amplification of MET and/or other genes involved in cell survival. Coordination of tumor metabolism and autophagy is also critical for resistance to HGF/MET-targeted therapy [24]. PF02341066 successfully restrained the malignant properties of MNNG/HOS cells in vitro as well as the growth and osteolysis/matrix production of orthotopic xenografts in vivo [25]. Therefore, PF02341066 might be a potential effective targeted therapy for OS patients. However, the effect and potential limitations of targeting c-Met in OS are still unclear. Considering the complex heterogeneity of c-Met expression in OS patients, we utilized two human OS cell lines with different malignant phenotypes and genetic statuses of c-Met to explore the therapeutic potential of the c-Met inhibitor, including MG-63 expressing wild-type MET and MNNG/HOS with TPR-MET rearrangement, a form of constitutive activation that is independent of ligands [26]. Here, we investigated the effect of OS heterogeneity on the efficacy of PF02341066 and revealed the contribution of the vascular niche in acquired resistance to PF02341066.

## 2. Materials and Methods

### 2.1. Cell Culture

Human OS cell lines MNNG/HOS (RRID: CVCL_0439) and MG-63 (RRID: CVCL_0426) were purchased from the Cell Bank of the Chinese Academy of Sciences (Shanghai, China) and authenticated using STR profiling within the last three years. Both cells were cultured in DMEM-F12 (DF12) medium (Sigma-Aldrich, St. Louis, MO, USA,) supplemented with 5% FBS (Biological Industries, Kibbutz Beit Haemek, Israel) or in DF12 containing 10 μg/mL human insulin, 5 μg/mL human transferrin, 10 μM 2-aminoethanol, 10 nM sodium selenite and 10 μM mercaptoethanol for serum-free culture. HUVECs were harvested from umbilical cord veins after collagenase treatment and cultured on plates coated with collagen I in Human Endothelial-SFM medium (Life Technologies, Gaithersburg, MD, USA) containing 10% FBS and 15 μg/mL endothelial cell growth supplement (Merck Millipore, Darmstadt, Germany). Low-passaged HUVECs (4 to 7 passages) were used for stable cell conditions in this study.

The MNNG/HOS-OSCs were obtained as previously described [11]. When MNNG/HOS cells were cultured in serum-free DF medium supplemented with 10 μg/mL human insulin, 5 μg/mL human transferrin, 10 μM 2-aminoethanol, 10 nM sodium selenite, and 10 μM mercaptoethanol, a portion of this monolayer of cells gradually changed to form sarcospheres, which present the characteristics of osteosarcoma stem cells (OSCs). OSCs were separated from the adherent cells and suspended in the culture medium.

All the cells were maintained at 37 °C in a humidified atmosphere containing 5% CO_2_ and confirmed to be free from microbiological contamination and cross-contamination. All experiments were performed with mycoplasma-free cells.

### 2.2. Colony Forming Assay

Low-melting-point agar was melted and mixed with serum-free DF12 media at a 1:1 ratio to make a supporting bottom layer in a 6-well plate (1.5 mL/well). The bottom agar layer was allowed to solidify at room temperature for 30 min. Three thousand single living MNNG/HOS-OSCs suspended in serum-free medium with 1:500 B27 were mixed with 0.35% soft agar and laid on top of the supporting layer. After incubation for 24 h, 1 mL medium supplemented with 0.01% DMSO or 0.05 μM PF02341066 for each group was added. Soft agar cultures were incubated for 14 days and fed with 1 mL fresh medium every three days. The colonies were fixed with 4% PFA for 30 min at room temperature and stained with crystal violet (Sigma-Aldrich, St. Louis, MO, USA). More than three wells were prepared for each condition, and the colony formation ability was assessed by counting the number of colonies.

### 2.3. VM Formation

Forty μL of Matrigel (Corning, NY, USA) was added to each well of a 96-well plate and allowed to polymerize for 1 h at 37 °C. MNNG/HOS-OSCs cells (1 × 10^4^ cells/well) were added to the Matrigel-coated wells in 100 μL DF12 supplemented with 5% FBS. The cells were then maintained at 37 °C in a humidified atmosphere containing 5% CO_2_ for 96 h to form VM, as previously described [11].

### 2.4. In Vitro Co-Culture System of MNNG/HOS and HUVEC

MNNG/HOS cells (4 × 10^4^) were seeded in a 6-well plate and HUVEC cells (5 × 10^4^) were seeded in a transwell chamber, respectively. The next day, the chamber seeded with HUVEC cells was transferred into the 6-well plate seeded with MNNG/HOS cells. Both cells were then co-cultured in a Human Endothelial-SFM medium added with 10% FBS and 15 μg/mL endothelial cell growth supplements.

### 2.5. RNA Isolation and Quantitative Real-Time PCR (qRT-PCR)

Total RNA was isolated using Trizol reagent (Magen, Guangzhou, China) and reverse transcribed using First Strand cDNA Synthesis Kit ReverTra Ace-α (TOYOBO, Osaka, Japan) according to the manufacturer’s recommendations. A 10 μL PCR-amplified reaction was performed according to the manufacturer’s instructions of LightCycler^®^ 480 SYBR Green I Master (Roche, Basel, Switzerland). All experiments were performed in triplicate. The sequences of the primers used in this study were as follows: MET forward, 5′- GGAGCCAAAGT-CCTTTCATCTGTAA-3′; MET reverse, 5′- GCAATGGATGATCTGGGAAATAAGAAG-AAT-3′; VEGFR2 forward, 5′- GGACTCTCTCTGCCTACCTCAC-3′; VEGFR2 reverse, 5′-GGCTCTTTCGCTTACTGTTCTG-3′; and GAPDH forward, 5′-GGAGCGAGATCCCTC-CAAAAT-3′; GAPDH reverse, 5′-GGCTGTTGTCATACTTCTCATGG-3′. The calculation of relative change in mRNA was performed using melting curve analysis and normalized to the housekeeping gene GAPDH.

### 2.6. Western Blotting Analysis

MNNG/HOS and MG-63 cells were seeded in a 6-well plate (1 × 10^5^ cells/well) and cultured in a serum-free medium. To detect the effect of PF02341066 (Selleck, Houston, TX, USA) on the activity of c-Met, OS cells were treated with 0 μM, 0.05 μM, 0.1 μM and 0.5 μM PF02341066 for 48 h. Cell lysates were then harvested and subjected to 8 or 10% SDS-PAGE as previously described [10]. Antibodies purchased from Cell Signaling Technology (CST, Danvers, MA, USA) for Western blots were as follows: c-Met, phospho-c-Met, VEGFR2, phospho-VEGFR2 and GAPDH.

### 2.7. Co-Immunoprecipitation (co-IP)

The cells (1 × 10^7^) were rinsed twice with cold phosphate buffered saline (PBS) and then extracted with 1 mL lysis buffer (50 mM pH 7.5 Tris, 150 mM NaCl, 0.5% Triton X-100, 10% glycerol, 2 mM EDTA, freshly added 1 mM PMSF, 10 mM N-ethylmaleimide, 20 mM each protease inhibitor cocktail) at 4 °C for 30 min. The cell lysates were then centrifuged at 14,000× *g* at 4 °C for 15 min. An amount of 200 μL of the collected supernatants was then incubated with 30 μL beads precoated with 5 μg control IgG (CST, Danvers, MA, USA) or 5 μg anti-MET (CST, Danvers, MA, USA) or anti-VEGFR2 (CST, Danvers, MA, USA) antibody at 4 °C overnight. The beads were then sequentially washed 5 times with co-IP-lysis buffer. The bound proteins were eluted with 50 μL 2% SDS buffer and boiled at 100 °C for 10 min. The proteins were then analyzed by Western blotting.

### 2.8. Cell Viability Assay

After 72 h co-culture with HUVEC, MNNG/HOS and co-MNNG/HOS cells were seeded at 5 × 10^3^ cells per well in 96-well plates. When cells were adherent and had morphologically spread, cells were treated with the indicated concentration of XL184 (Selleck, Houston, TX, USA). After 48 h, 10 μL CCK-8 solution (Dojindo Molecular Technologies, Kumamoto, Japan) was added to each well and incubated for 2 h at 37 °C in a humidified incubator. The light absorption was measured at 450 nm with a microplate reader.

### 2.9. TUNEL Apoptosis Assay

OS cells cultured in serum-free medium were treated with 0.05 μM and 0.5 μM PF02341066 for 24 h and fixed in 4% paraformaldehyde (PFA) for 30 min at room temperature. TUNEL staining was performed according to the manufacturer’s protocol of In Situ Cell Death Detection Kit, POD (Roche, Basel, Switzerland) and then visualized by fluorescence microscopy (Carl Zeiss, Jena, Germany). The quantification of the TUNEL assay was performed by counting the total number of cells (based on DAPI signal) and the number of TUNEL positive cells. Results were expressed as TUNEL-staining cells/total cells.

### 2.10. Vector Construction and Cell Transfection

To obtain stable MNNG/HOS cells that overexpress luciferase, the luciferase gene fragment containing NotI and AscI double restriction sites was amplified from the pGL3-control vector (Promega, Madison, WI, USA). Luciferase fragment was ligated into entry vector and then inserted into plX302 vector (Addgene, Watertown, MA, USA) by LR reaction, followed by screening and the selection of positive clones with ampicillin as previously described [27]. The primer sequences used for luciferase gene fragment were as follows: forward, 5′-TTGCGGCCGCATGGAAGACGCCAAAAACATAAA-3′; reverse, 5′-TTGGCGCGCCTTACACGGCGATCTTTCCGCC-3′.

### 2.11. Animal Experiments

All animal experiments were conducted according to the institutional and national guidelines for the care and use of laboratory animals with approval by the Animal Research Committee of Sun Yat-sen University. Four to six-week-old female BALB/c nude mice were purchased from the Laboratory Animal Center of Sun Yat-sen University and maintained in a pathogen-free facility with a 12 h light-dark cycle at a constant temperature (23 ± 0.5 °C). MNNG/HOS cell suspension (4 × 10^6^ single cells) in 100 μL PBS was intramuscularly injected into the leg muscle of each mouse. When tumors of approximately 0.5 cm in diameter formed, mice were treated with vehicle, 50 mg/kg PF02341066 and 50 mg/kg XL184 by oral gavage every other day. Tumor length and width were measured using a vernier caliper, and the tumor volumes were calculated as follows: volume (cm^3^) = length × width^2^/2. After treatment, mice were analyzed by in vivo bioluminescence IVIS imaging (IVIS Lumina XR Series III). Fifteen minutes prior to imaging, mice were IP injected with 150 mg/kg D-Luciferin (PerkinElmer, Waltham, MA, USA). Tumor tissues were then used for ex vivo bioluminescence imaging and histological analysis.

### 2.12. Hematoxylin-Eosin (H&E), Periodic Acid-Schiff (PAS) Staining and Immunohistochemistry (IHC)

The tissues were fixed in 10% buffered formalin for 24 h, dehydrated in ethanol, embedded in paraffin and then sliced into 5 μm-thick sections. The tissue sections were stained with H&E or PAS staining Kit (Sigma-Aldrich, St. Louis, MO, USA) according to the manufacturer’s instructions.

For IHC analysis, the sections were incubated with primary antibody overnight at 4 °C, and were then incubated with secondary antibody for 1 h at room temperature. Nuclei were stained with hematoxylin. Staining signals of IHC were visualized using MaxvisionTM3 HRP-Polymer (Mouse/Rabbit) IHC Kit (Maxim Biotechnologies, Rockville, MD, USA). Antibodies against c-Met (Cat. No. #8198), phospho-c-Met (Cat. No. #3077), VEGFR2 (Cat. No. #2479) and phospho-VEGFR2 (Cat. No. #2478) purchased from CST, CD31 (Cat. No. MA5-13188) from Thermo Fisher Scientific and NuMA (Cat. No. ab84680) from Abcam were used for IHC staining. For quantitative analysis, five random fields of each stained section were captured under a light microscope. Image scoring analysis was performed using the IHC profiler in Image J (Version 7.0, Media Cybernetics, Bethesda, MD, USA). The proportions of cells with weak staining, moderate staining, and strong staining in each image were counted separately. Quantification of Ki67, p-Met and p-VEGFR2 staining for each sample was determined by histoscore (H-score) calculated as follows: 1 × percentage of low positive + 2 × percentage of positive + 3 × percentage of high positive [28].

For CD31-PAS dual staining, CD31 immunohistochemical staining was performed as per the IHC procedures and finally with the PAS staining Kit (Sigma-Aldrich, Cat. No. 101646) according to the manufacturer’s instructions. The slides were viewed under a microscope to detect the area of the vessels. Quantification of CD31-positive vessels and PAS-staining VM was analyzed by Image J (Version 7.0, Media Cybernetics, Bethesda, MD, USA). Images of HPF were captured randomly from each slide and color segmentation was used to identify CD31-positive and PAS-staining areas. The percentage of the vessel area was determined by the areas of CD31-positive and PAS-staining divided by the total area [29].

### 2.13. Immunofluorescence

Immunofluorescent staining was performed on the slides coated with monolayer cells. After being fixed in 4% PFA and permeabilized in 0.2% Triton X-100, the cells were blocked in 10% normal goat serum for 1 h and then incubated with primary antibody at 4 °C overnight. The slides of the mounting cells were subsequently incubated with Alexa Fluor 488 secondary antibody (Invitrogen, Carlsbad, CA, USA) for 1 h at room temperature. After being counter stained with DAPI (Invitrogen, Carlsbad, CA, USA) for 5 min, the slides were observed under a fluorescent microscope. For MNNG/HOS-OSCs, the cells were collected and centrifuged at 1500 rpm for 5 min, and cell pellets were fixed in 4% PFA followed by immunofluorescent staining procedures. After being counter stained with DAPI, the cells were suspended and mounted using a fluorescence mounting medium (Dako, Glostrup, Denmark) on the slide. Images were acquired using a fluorescence microscope.

### 2.14. Statistical Analysis

All the experiments were performed at least in triplicate independently. Values were presented as mean ± standard error. Statistical significance was determined by using the two-tailed Student’s *t*-test, one-way ANOVA or two-way ANOVA on GraphPad Prism (Version 9.0, GraphPad Software, San Diego, CA, USA). *p* values < 0.05 were considered significant.

## 3. Results

### 3.1. PF02341066 Selectively Inhibited the Proliferation and Survival of OS Cells

To evaluate the tumor suppressive effect of c-Met inhibition, MNNG/HOS and MG-63 cells were respectively cultured in serum-free medium and treated with various concentrations of the c-Met inhibitor PF02341066 for 48 h. Compared with the control group, the proliferation of MNNG/HOS cells was remarkably reduced after PF02341066 treatment in a dose-dependent manner (IC50 = 0.04 μM, Figure 1a). However, PF02341066 had a weaker inhibitory effect on the proliferation of MG-63 cells (IC50 > 1 μM, Figure 1a). Western blotting analysis showed that after being treated with increasing concentrations of PF02341066, c-Met phosphorylation was completely abrogated (Figure 1b). In addition, a notable pro-apoptotic effect was observed in MNNG/HOS cells after being treated with PF023412066 for 24 h. The TUNEL assay indicated that the number of apoptotic MNNG/HOS cells dramatically increased as the concentration of PF02341066 increased (Figure 1c,d). In contrast, neither 0.05 μM nor 0.5 μM PF02341066 treatment induced apoptosis of MG-63 cells (Figure 1e,f). These results demonstrated that the c-Met inhibitor had a specific cell-killing effect on MNNG/HOS with high phosphorylation of c-Met but displayed an unsatisfactory effect on MG-63 cells with low c-Met phosphorylation. We reasoned that the different responses to c-Met targeted inhibition between cell lines might be caused by the different forms of c-Met activation. PF02341066 is ideal for treating OS patients who are dependent on constitutively activated c-Met signaling.

### 3.2. c-Met Inhibitor Suppressed the Ability of Self-Renewal and Survival of MNNG/HOS-OSCs

CSCs in OS have been identified as a subpopulation that is associated with drug resistance and recurrence [30]. Based on our previous study, microenvironmental signals could endow adherent OS cells with stemness and promote the formation of OSCs [11]. Immunofluorescence (Figure 2a) and Western blotting analysis (Figure 2b) showed that the MNNG/HOS-OSCs presented elevated levels of c-Met phosphorylation compared with adherent MNNG/HOS cells. PF02341066 treatment efficiently blocked the phosphorylation of c-Met in MNNG/HOS-OSCs (Figure 2a,b). Soft agar colony formation assay demonstrated that MNNG/HOS-OSCs treated with PF02341066 formed less colonies than the DMSO-treated cells (Figure 2c,d). Moreover, PF02341066 treatment significantly reduced cell viability (Figure 2e,f) and promoted the apoptosis of MNNG/HOS-OSCs (Figure 2g,h). Taken together, these results confirmed that c-Met inhibitor effectively inhibited the proliferation and survival of MNNG/HOS-OSCs.

### 3.3. PF02341066 Selectively Inhibited the Proliferation and Survival of OS Cells

Because the in vitro experiments revealed that the c-Met inhibitor PF02341066 effectively suppressed the proliferation and induced apoptosis of MNNG/HOS cells and MNNG/HOS-OSCs, we then investigated whether PF02341066 could reduce tumor growth in vivo. MNNG/HOS cells were intramuscularly transplanted into BALB/c nude mice. On day 6, a tumor of approximately 0.5 cm in diameter formed, and mice were treated with PF02341066 (50 mg/kg), as illustrated in Figure 3a. During the first 9 days of treatment, tumor growth was significantly suppressed in the PF02341066 group compared to the vehicle group (Figure 3b). IHC analysis showed that c-Met phosphorylation was dramatically reduced in xenografts after PF02341066 treatment (Figure 3c,e). However, tumors in the PF02341066 group showed abrupt growth after 10 days of treatment (Figure 3b). Ki67 expression in the PF02341066 group was strikingly higher than that in the vehicle group, suggesting that PF02341066 might somehow induce MNNG/HOS cell proliferation in the late stage of drug administration (Figure 3d). Additionally, quantitative analyses on the expression of p-Met and Ki67 are indicated in Figure 3f. We then performed a drug withdrawal experiment according to the timeline (Figure 3g). After withdrawing PF02341066 treatment on day 14, the tumors from both the PF02341066 and vehicle groups grew to approximately the same size (Figure 3h,i). Thus, PF02341066 could not inhibit MNNG/HOS cell proliferation in vivo as efficiently as in vitro and might cause a rebound of tumor growth through an unidentified compensatory mechanism.

### 3.4. PF02341066 Selectively Inhibited the Proliferation and Survival of OS Cells

OS is a highly vascularized tumor characterized by abundant microvessels and VM. Patients with lower MVD show a better response to neoadjuvant chemotherapy and a higher survival rate [31]. Interestingly, more vessels (Figure 4a,b) and less necrosis (Figure 4c,d) were detected in xenografts after PF02341066 treatment. Compared to the vehicle group, IHC staining further revealed an obvious downregulation of c-Met phosphorylation accompanied by the activation of VEGFR2 in OS cells adjacent to the vascular microenvironment, which was indicated by CD31-positive staining in the PF02341066 treatment group (Figure 4e,f). Additionally, PAS staining showed that OS cells could form VM. Similarly, OS cells adjacent to VM had less c-Met phosphorylation (Figure 4g) and higher VEGFR2 activation (Figure 4h) than those far from VM. We postulated that drug-sensitive OS cells might escape the PF02341066-induced proliferative inhibition and apoptosis by virtue of the vascular niche.

To confirm this speculation, MNNG/HOS cells were co-cultured with HUVECs in a transwell system (co-MNNG/HOS) to mimic OS cells adjacent to tumor vessels. In line with the findings of the animal experiment, the effect of PF02341066 on MNNG/HOS cells in proliferation inhibition (Figure 4i,j) and apoptosis induction (Figure 4k,l) was dramatically reduced in co-MNNG/HOS cells. MNNG/HOS-OSCs possessed the capability to transdifferentiate into CD31-positive endothelial-like cells and construct a vascular network [11]. PF02341066 treatment on co-MNNG/HOS and VM also failed to induce cell apoptosis, as determined by TUNEL apoptosis staining (Figure 4k,l). Therefore, we speculated that vascular endothelial cells or OS cells with VM status could restrain c-Met activation and block the efficacy of c-Met inhibitor treatment.

### 3.5. PF02341066 Selectively Inhibited the Proliferation and Survival of OS Cells

Active angiogenesis and intensive VEGFR2 expression in PF02341066-treated xenografts prompted us to clarify the potential role of VEGFR2 during c-Met inhibitor treatment and elucidate the interaction between c-Met and VEGFR2. The expression and phosphorylation of c-Met were significantly inhibited in co-MNNG/HOS cells, whereas expression and phosphorylation of VEGFR2 were significantly increased as determined by qRT-PCR (Figure 5a,b) and Western blotting assay (Figure 5c). Immunofluorescent staining revealed that the phosphorylation of c-Met was abrogated after MNNG/HOS-OSCs formed VM (Figure 5d). Notably, VEGFR2 signaling in both co-MNNG/HOS cells (Figure 5c) and VM channels formed by MNNG/HOS-OSCs cells (Figure 5e) was activated, which is consistent with the in vivo results (Figure 4f,h).

VEGFA is the most potent angiogenic promoting factor that mediates the activation of the VEGFR2 [32] and the inhibition of the c-Met signaling [33]. Western blotting assay indicated that VEGFA inhibited c-Met phosphorylation in MNNG/HOS cells (Figure 5f). Immunoprecipitation assay further revealed a physical interaction between c-Met and VEGFR2, suggesting that ligands of these two receptors might be redistributed into a c-Met/VEGFR2 complex (Figure 5g). Hence, these data supported the notion that MNNG/HOS cells became independent of c-Met signaling and acquired the capacity to withstand PF02341066-induced apoptosis and survive through VEGFR2 activation. VEGFR2 activation in MNNG/HOS cells might confer resistance to c-Met-targeted therapy as a compensatory pathway for c-Met inhibition.

### 3.6. PF02341066 Selectively Inhibited the Proliferation and Survival of OS Cells

Based on the above in vitro and in vivo observations, we further explored whether dual targeting c-Met and VEGFR2 could exert a lasting cytocidal effect on OS cells. XL184 (cabozantinib) is an ATP-competitive inhibitor of a wide range of kinase receptors that most potently inhibits both c-Met and VEGFR2 receptors [34]. Compared with the vehicle group, the proliferation of MNNG/HOS (IC50 = 0.057 μM) and co-MNNG/HOS (IC50 = 0.077 μM) cells was markedly repressed after XL184 treatment (Figure 6a). Western blotting analysis showed that XL184 treatment significantly inhibited the phosphorylation of c-Met and VEGFR2 in both MNNG/HOS and co-MNNG/HOS (Figure 6b). To investigate the effects of XL184 treatment on MNNG/HOS in vivo, MNNG/HOS cells stably overexpressing luciferase (MNNG/HOS-Luc+) were established for bioluminescence imaging. MNNG/HOS-Luc+ cells were intramuscularly transplanted into BALB/c nude mice. On day 6, a tumor of approximately 0.5 cm diameter formed, and the mice were treated with PF02341066 or XL184 (50 mg/kg), as illustrated in Figure 6c. Treatment with XL184 markedly suppressed the growth of tumors compared to either the vehicle- or PF02341066-treated groups (Figure 6d). The bioluminescence imaging analysis revealed that MNNG/HOS-Luc+ bearing mice treated with XL184 presented minimal signals (Figure 6e,f). This drug showed a higher potency of inhibition that both c-Met and VEGFR2 phosphorylation could be efficiently suppressed with this dual targeting inhibitor treatment than the vehicle group (Figure 6g,h). Furthermore, we performed IHC analysis on xenograft tissues using the cell proliferation-associated marker Ki67 to examine the growth-inhibitory effect of XL184. The Ki67 nuclear expression and intensity in the XL184 group were much lower than those in the other groups, suggesting that XL184 successfully suppressed the proliferation of MNNG/HOS in vivo (Figure 6i). Besides, quantitative analyses on the expression of p-Met, p-VEGFR2 and Ki67 are indicated in Figure 6j–l. In summary, the above results collectively demonstrated that dual targeting c-Met and VEGFR2 could be a promising therapeutic strategy for OS patients in future clinical development.

## 4. Discussion

Despite great advances achieved in the novel paradigm of cancer-targeted therapy, various obstacles still remain and need to be discussed and addressed. Numerous clinical trials aiming to improve treatment in the past decades have failed to achieve higher cure rates [35]. Drug resistance has long been recognized as a hallmark of malignant cancer and represents an ongoing challenge in the development of personalized cancer medicine. Drug resistance can be divided into two categories, intrinsic and adaptive, which manifest as primary and acquired drug resistance, respectively [36]. Acquired drug resistance not only hinders the duration of tumor response but also limits long-term survival in genotype-matched precision medicine [37,38].

Aberrant c-Met signaling has been implicated in multiple cancer types and is crucial for tumor development and progression such as stemness, proliferation, survival and angiogenesis [39,40,41]. c-Met has attracted attention as a promising target in various cancers, such as non-small-cell lung cancer with MET alteration, MET-amplified esogastric adenocarcinomas and c-Met-overactive colorectal cancer [42,43]. Clinically, drug resistance to c-Met-targeted therapy develops later while receiving therapy despite the initial striking tumor regression, resulting in therapeutic failure and ultimate patient death. There are some widely accepted mechanisms responsible for the emergence of acquired drug-resistant clones in the course of targeted therapy. In this study, we highlighted various aspects of the heterogeneity of OS that might interfere with c-Met-targeted treatment susceptibility and confer resistance, and we strived to provide a basis for rational therapy improvement.

To this end, MNNG/HOS and MG-63 cells that represented different subtypes were employed to evaluate the efficacy of the c-Met inhibitor PF02341066 on OS. These OS cell lines differ in levels of c-Met expression and regulation. In this study, we showed that PF02341066 specifically killed OS cells with active c-Met phosphorylation. Therefore, recognizing the inter-heterogeneity of c-Met activity among OS patients and choosing treatment-sensitive patients for targeted therapy can hopefully deliver the optimal therapeutic effect.

Our previous results have indicated that a small subpopulation of OS cells with stem cell-like properties are responsible for therapy resistance and recurrence in OS [10,11]. Considering that OSCs are a recalcitrant source of drug resistance, the influences of c-Met inhibitor PF02341066 on OSCs were further examined in this work. OSCs derived from sensitive OS cells were responsive to PF02341066, as their survival and self-renewal were significantly inhibited. Thus, PF0234106 had a robust cytocidal effect on both sensitive OS cells and OSCs. However, despite the initial remarkable tumor regression, c-Met-targeted therapy of sensitive OS failed to inhibit tumor growth in an orthotopic xenograft model and even demonstrated a rebound after drug withdrawal.

At present, most studies that focus on the mechanisms of acquired drug resistance of cancer cells can be divided into two major categories. One mechanism of acquired drug resistance is the mutation and amplification of MET. Studies have indicated that the Y1230 and D1228 mutations and increased MET copy number may be common mechanisms of resistance to c-Met inhibitors [24]. In addition, the functional alteration of signaling molecules, such as the mutation of KRAS, HRAS or EGFR, and the amplification of HER2 are also involved in therapeutic resistance to c-Met-targeted drugs [44]. Our results found drug resistance during the late phase of treatment, and the p-Met levels in OS cells were still reduced at this time, suggesting other potential causes.

There is also evidence that the microenvironment can endow cancers with the ability to withstand targeted drugs. It has been shown that lactate enhances the formation of HGF by tumor-associated fibroblasts, which in turn activates c-Met-dependent signaling pathways in cancer cells and confers resistance to c-Met inhibitors [45]. In this report, we observed that angiogenesis became more vigorous after PF2341066 treatment. When OS cells were adjacent to vascular endothelial cells or formed VM, c-Met expression and phosphorylation were dramatically reduced, allowing OS cells to tolerate the c-Met inhibitor.

In a cancer microenvironment, the vascular niche can regulate cancer cells by secreting growth factors, cytokines or exosomes [46,47]. There are a number of secretome factors such as VEGFA, CXCL1, IL8, PDGFRβ, BMP6, ANGPT2 and other factors in the neovascular niche. As the most potent angiogenic promoting factor which mediates the activation of VEGFR2 [32], VEGFA was found to slightly upregulate the expression of VEGFR2 in MNNG/HOS. It seems that VEGFA is one of the factors that could regulate the expression of c-Met and VEGFR2 in OS cells. In addition, we also found that the use of a DNA methylation inhibitor (5-azadCyd) could restore the downregulated expression of c-Met after co-culture with HUVEC, suggesting that DNA methylation may play a role in the downregulation of c-Met. In conclusion, the vascular microenvironment could downregulate c-Met expression and inhibit its phosphorylation activity, but it could also upregulate VEGFR2 expression and activate phosphorylation in sensitive OS cells, thereby protecting them from apoptosis which is induced by the inhibition of c-Met phosphorylation, thus leading to the failure of c-Met-targeted therapy.

There is evidence that cancer-induced blood vessels are different from normal blood vessels, which leads to the failure of traditional antiangiogenic therapies such as classical anti-VEGF therapy [48]. In addition, the VM channels formed by cancer cells are actively involved in the tumor vasculature, which desensitizes tumors to existing antiangiogenic drugs [49]. Targeting VEGFR2 can effectively reduce the angiogenesis of cancer tissues, thereby cutting off the nutritional supply of tumors and inhibiting tumor growth [50]. Recent studies have also shown that the abnormal activation of VEGFR2 in tumors is associated with poor prognosis [51].

The VEGF-VEGFR2 signaling pathway is known as the classical angiogenesis-related pathway and has been reported to negatively regulate the c-Met activity. We found that OS cells adjacent to the vascular niche or formed VM had a significant activation of VEGFR2, which might play a compensatory role in replacing c-Met dependence in OS cells. We also detected a direct interaction between VEGFR2 and c-Met. Studies have revealed VEGFR2 to be a negative regulator of other RTKs by blocking their activation through direct interaction. VEGFR2 directly and negatively regulates c-Met through enhanced recruitment of protein tyrosine phosphatase 1B (PTP1B) to a c-MET/VEGFR2 heterocomplex [33]. Our animal experiment results showed that dual-target inhibition effectively alleviated the drug resistance caused by single-targeted c-Met therapy. Consistent with our in vivo results, dual targeting c-Met and VEGFR2 inhibited the growth and metastasis of the hepatocellular carcinoma [52]. Hence, these results demonstrated that the vascular niche played a vital role in the acquired resistance to c-Met-targeted therapy.

## 5. Conclusions

In summary, the heterogenous nature of OS accompanied by acquired and intrinsic resistance to targeted therapy can account for stagnation in clinical improvement. Exploring the relationship between the vascular niche and tumor cells will shed light on OS research and clinical treatment. Specially targeted treatment strategies for individual OS patients based on a more detailed molecular and cellular classification are essential. Unveiling susceptibility and resistance regulated by heterogeneity might help improve clinical outcomes by both identifying patients who might be cured with less toxic interventions and also by targeting resistance mechanisms to sensitize OS to treatment.

## Figures and Tables

**Figure 1 cancers-14-06201-f001:**
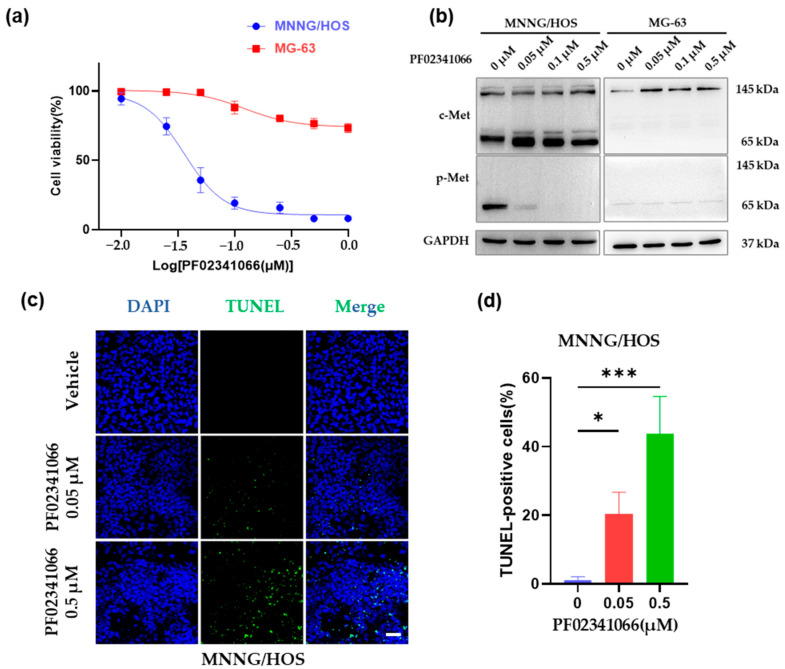
c-Met inhibitor PF02341066 showed a dose-dependent inhibition of OS cells proliferation and survival. (**a**) MNNG/HOS and MG-63 cells were cultured in the presence of the indicated concentrations of PF02341066 for 48 h prior to fixation and crystal violet staining. Crystal violet staining was quantified by solubilizing the fixed dye and assessing the absorbance at 570 nm. Values indicated were normalized to DMSO control. Mean ± SEM, three independent experiments, each with *n* = 3. The concentration of drug needed to reduce plaque numbers by half (IC50) was calculated by fitting the data to a linear regression model using Prism software (GraphPad Prism Software, Inc., La Jolla, CA, USA). (**b**) Western blotting analysis on the expression and the phosphorylation of c-Met in MNNG/HOS and MG-63 cells cultured in serum-free medium with the indicated concentrations of PF02341066 for 48 h. Detailed information about Western blot can be found at Appendix A. (**c**–**f**) Representative images of the TUNEL assay. MNNG/HOS and MG-63 cells were cultured in the presence of the indicated concentrations of PF02341066 for 24 h. Apoptosis was determined by TUNEL assay (green channel) in samples either treated with PF02341066 or DMSO for control. DAPI (blue channel) was used to locate the nuclei of the cells. Scale bar = 50 µm. (**d**,**f**) Quantitative analysis of TUNEL-positive MNNG/HOS and MG-63 cells. Mean ± SEM, three independent experiments, each with *n* = 3. *** *p* < 0.001, * *p* < 0.05, ns *p* > 0.05, by one-way ANOVA.

**Figure 2 cancers-14-06201-f002:**
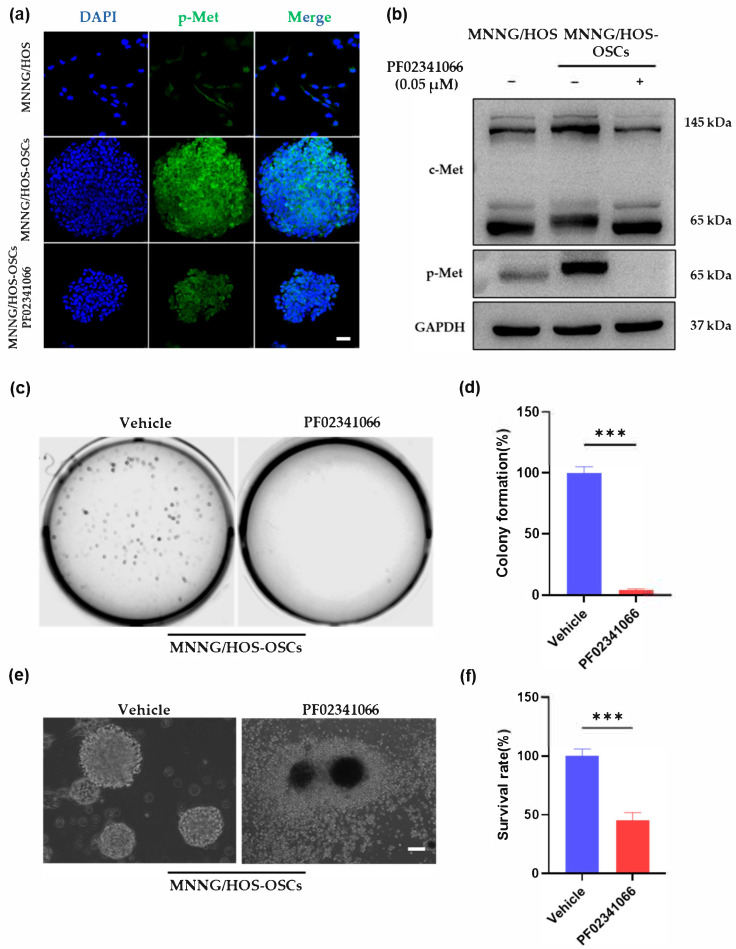
c-Met inhibitor could suppress the proliferation and survival of sensitive MNNG/HOS-OSCs. (**a**) Immunofluorescence staining of MNNG/HOS cells and MNNG/HOS-OSCs with DMSO or 0.05 μM PF02341066. Scale bar = 50 µm. (**b**) Western blotting analysis on the protein level of c-Met and its phosphorylation in MMNG/HOS-OSCs with DMSO or 0.05 μM PF02341066 treatment. Detailed information about Western blot can be found at Appendix A. (**c**,**d**) Representative images (**c**) and quantitative analysis (**d**) of soft agar colony formation assay of MNNG/HOS-OSCs treated with DMSO or 0.05 μM PF02341066. Mean ± SD, *n* = 3. *** *p* < 0.001 by two-tailed Student’s *t*-test. (**e**) Representative bright-field microscopy images showing the MNNG/HOS-OSCs in the serum-free medium after being treated with DMSO or 0.05 μM PF02341066 for 72 h. Scale bar = 100 µm. (**f**) Quantitative analysis of crystal violet assay for determining the viability of MNNG/HOS-OSCs after being treated with DMSO or 0.05 μM PF02341066 for 48 h. Mean ± SD, *n* = 3. *** *p* < 0.001 by two-tailed Student’s *t*-test. (**g**) TUNEL apoptosis staining of MNNG/HOS-OSCs treated with 0.05 μM PF02341066 for 48 h. TUNEL staining was green and nuclear staining (DAPI) was blue. Scale bar = 50 µm. (**h**) Quantitative analysis of TUNEL-positive MNNG/HOS-OSCs cells. Mean ± SEM, three independent experiments, each with *n* = 3. *** *p* < 0.001 by two-tailed Student’s *t*-test.

**Figure 3 cancers-14-06201-f003:**
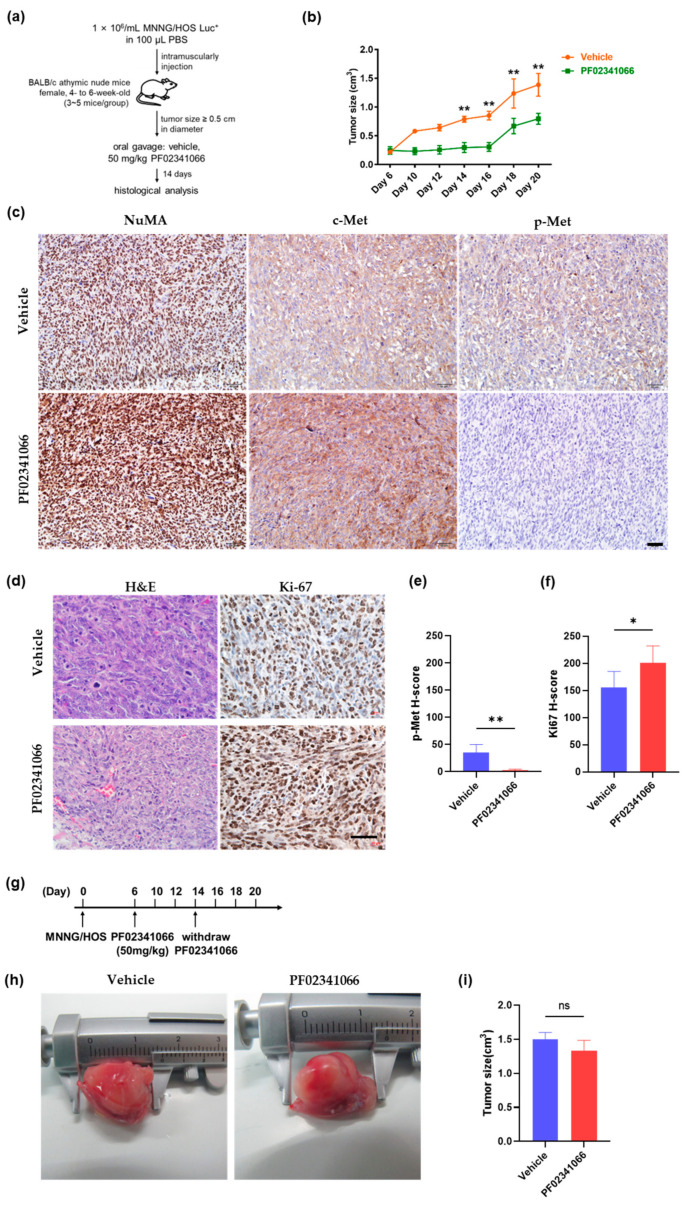
PF02341066 treatment of tumor-bearing mice in vivo. (**a**) Timeline for the in vivo animal experiment. Treatment started on day 6 when the tumors reached about 0.5 cm in largest diameter after tumor inoculation. Mice then received oral gavage of PF02341066 (50 mg/kg) for the next 2 weeks. (**b**) Quantification analysis on the volume of tumors. Mean ± SD, *n* = 5. ** *p* < 0.01 by two-tailed Student’s *t*-test. (**c**) The staining on c-Met and p-Met of tumor tissues from mice on day 20 after treatment. NuMA staining indicated human-derived MNNG/HOS cells. CD31 staining indicated tumor vessels. Scale bar = 50 µm. (**d**) H&E and Ki67 staining of tumor tissues from mice on day 20 after treatment. Ki67, a proliferating marker staining the nucleus. Scale bar = 50 µm. (**e**,**f**) Quantitative analysis on the expression of p-Met and Ki67. Mean ± SEM, five independent samples, each with *n* = 5. ** *p* < 0.01, * *p* < 0.05, ns *p* > 0.05 by two-tailed Student’s *t*-test. (**g**) Timeline for the drug withdrawal experiment. Treatment was withdrawn from day 14 according to the relapse timepoint. (**h**,**i**) Representative images (**h**) and statistical analysis (**i**) of tumors from mice on day 20 after withdrawing PF02341066. Mean ± SD, *n* = 5. ns *p* > 0.05 by two-tailed Student’s *t*-test.

**Figure 4 cancers-14-06201-f004:**
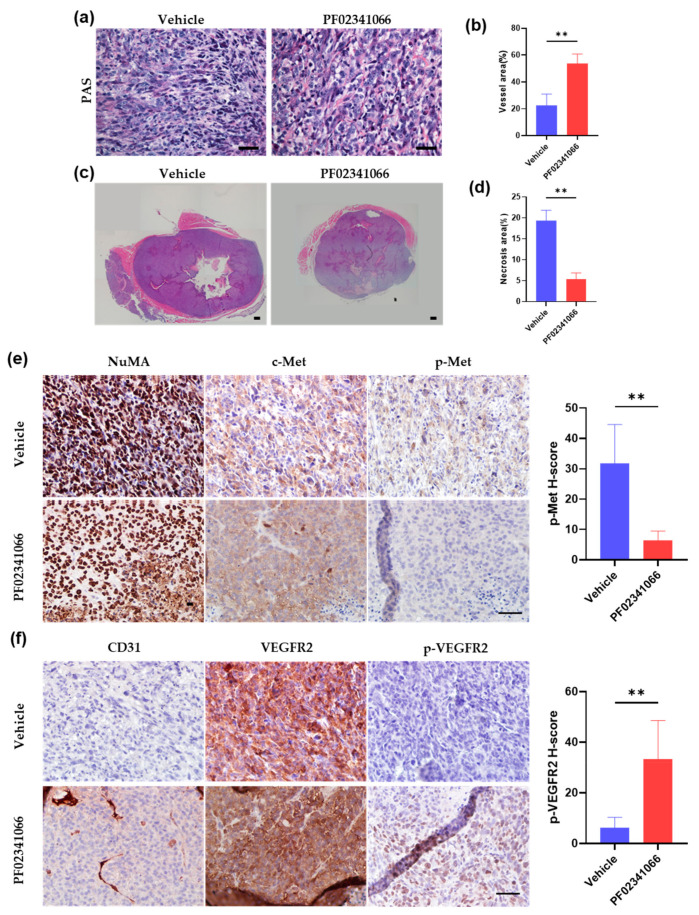
The vascular niche could promote drug resistance to PF02341066 treatment by reducing c-Met activation in MNNG/HOS cells. (**a**–**d**) Statistical analysis indicated the area of vessels (**a**,**b**) and necrosis (**c**,**d**) from xenograft tissues in the vehicle and PF02341066 group on day 20. Mean ± SEM, *n* = 3. ** *p* < 0.01 by two-tailed Student’s *t*-test. Scale bar = 50 µm. (**e**,**f**) IHC staining on c-Met and VEGFR2 phosphorylation of xenograft tissues from mice on day 20 after vehicle or PF02341066 treatment. NuMA staining indicated human-derived MNNG/HOS cells. CD31 staining indicated tumor vessels. Scale bar = 50 µm. Quantitative analysis of the expression of p-Met and p-VEGFR2. Mean ± SEM, five independent samples, each with *n* = 5. ** *p* < 0.01 by two-tailed Student’s *t*-test. (**g**) p-Met-PAS double staining of xenograft tissues from mice on day 20. Violet staining of PAS represented VM. Scale bar = 50 µm. (**h**) PAS and VEGFR2 or p-VEGFR2 staining of xenograft tissues from mice on day 20. Violet staining of PAS represented VM. Scale bar = 50 µm. (**i**,**j**) Respective bright-field microscopy images (**i**) and statistical analysis (**j**) showing MNNG/HOS and co-MNNG/HOS living cells after being treated with 0.05 μM PF02341066 for 48 h. Scale bar = 200 µm. Mean ± SD, *n* = 3. * *p* < 0.05 by two-tailed Student’s *t*-test. (**k**,**l**) Respective images. Scale bar = 50 µm. (**k**) and quantitative analysis (**l**) of TUNEL apoptosis assay on MNNG/HOS cells, co-MNNG/HOS cells and MNNG/HOS-VM after being treated with 0.05 μM PF02341066 for 48 h. Mean ± SEM, three independent experiments, each with *n* = 3. *** *p* < 0.001, ns *p* > 0.05 by two-way ANOVA.

**Figure 5 cancers-14-06201-f005:**
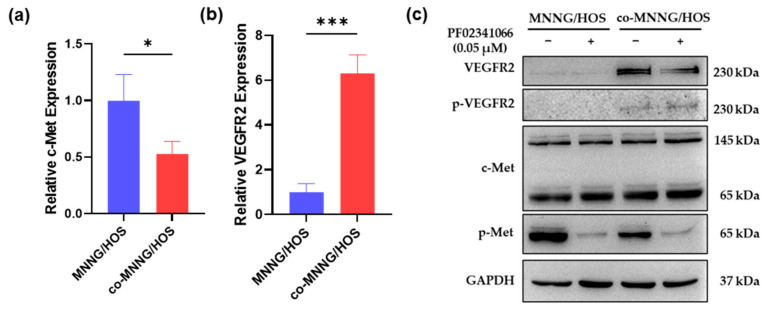
VEGFR2 activation might tie up with the downregulation of c-Met signaling in MNNG/HOS cells. (**a**–**c**) qRT-PCR (**a**,**b**) and Western blotting (**c**) analysis on the expression and phosphorylation of c-Met and VEGFR2 in MNNG/HOS cells and co-MNNG/HOS cells after being treated with 0.05 μM PF02341066 for 48 h. Mean ± SEM, three independent experiments, *n* = 3. *** *p* < 0.001, * *p* < 0.05 two-tailed Student’s *t*-test. (**d**) Immunofluorescence staining of MNNG/HOS-OSCs and VM revealed c-Met and the phosphorylation of c-Met. Scale bar = 50 µm. (**e**) IHC analysis on the activation of VEGFR2 in VM. Scale bar = 50 µm. (**f**) Western blotting analysis on the expression and the phosphorylation of c-Met in MNNG/HOS cells after 50 ng/mL VEGFA induction for 72 h. (**g**) Representative images of immunoprecipitation revealed a physical interaction between c-Met and VEGFR2. Detailed information about Western blot can be found at Appendix A.

**Figure 6 cancers-14-06201-f006:**
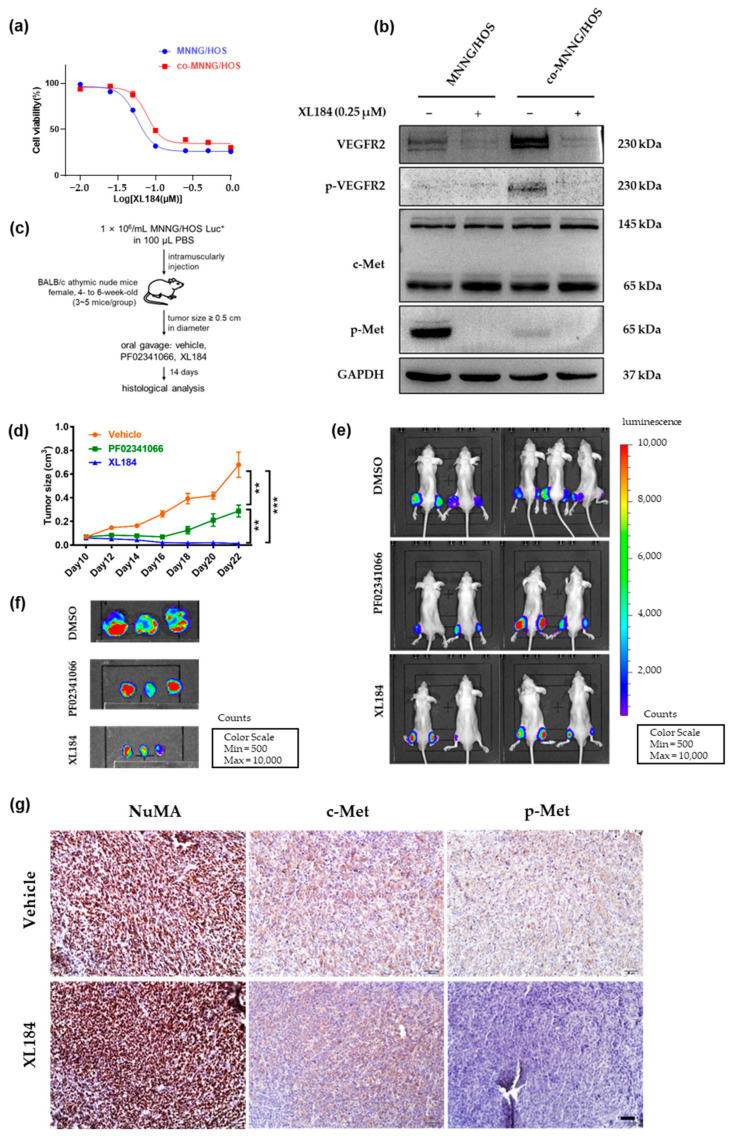
In vivo efficacy of XL184 in MNNG/HOS Luc+ xenograft model. (**a**) CCK8 assay revealed the proliferation of MNNG/HOS cells which were cultured in the presence of the indicated concentrations of XL184 for 48 h. (**b**) Western blotting analysis on the expression and the phosphorylation of c-Met and VEGFR2 in MNNG/HOS cells cultured in serum-free medium with the indicated concentrations of XL184 for 48 h. Detailed information about Western blot can be found at Appendix A. (**c**) Treatment schema. Treatment started on day 10 when the tumors reached about 0.5 cm in largest diameter after tumor inoculation. Mice then received oral gavage of PF02341066 and XL184 (50 mg/kg) every other day. Tumor growth was measured using a vernier caliper during 2-week treatment and monitored by bioluminescence imaging. (**d**) The tumor size of each mouse was measured and performed as the tumor volume (cm^3^) on the indicated day. Mean ± SD, *n* = 5. ** *p* < 0.01, *** *p* < 0.001 by one-way ANOVA. (**e**) In vivo bioluminescence imaging of BALB/c athymic nude mice bearing MNNG/HOS-Luc+ xenografts on day 22. (**f**) Representative images of MNNG/HOS-Luc+ xenografts that were harvested for ex vivo bioluminescence imaging. (**g**,**h**) c-Met and VEGFR2 activation of tumor tissues from mice on day 22 after treatment. (**i**) Ki67 staining activity of tumor tissues from mice under different treatments. Ki67, a proliferating marker staining the nucleus. NuMA staining indicated human-derived MNNG/HOS cells. CD31 staining indicated tumor vessels. Scale bar = 50 µm. (**j**–**l**) Quantitative analysis of the expression of p-Met, p-VEGFR2 and Ki67. Mean ± SEM, five independent samples, each with *n* = 5. ** *p* < 0.01, * *p* < 0.05, ns *p* > 0.05 by two-tailed Student’s *t*-test and one-way ANOVA.

## Data Availability

The data presented in this study are available on request from the corresponding author.

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
