# Peer review of "Vascular Niche Facilitates Acquired Drug Resistance to c-Met Inhibitor in Originally Sensitive Osteosarcoma Cells"

_cancers, 2022, doi:10.3390/cancers14246201_

Round 1

Reviewer 1 Report

The paper is well designed and presented, I have however three remarks:

Fig 1a and 1b, the linear regression instead bars should be presented.

Fig 6a also linear regression should be presented, why IC50 was not calculated from this chart?

Fig 6f the colour scale for the photo counts should be the same for DMSO, PF10231066 and XL184, the best could be if Fig 6e has the same scale.

Author Response

Response to Reviewer 1 Comments

Comment 1:

Fig 1a and 1b, the linear regression instead bars should be presented.

Response1:

Thank you for your suggestion. We have replaced the bar plot with a linear regression plot in Figure 1a.

Comment 2:

Fig 6a also linear regression should be presented, why IC50 was not calculated from this chart?

Response 2

Thank you for noticing. We have changed the bars to a linear regression plot and added IC50 values in page 17, line 454.

Comment 3:

Fig 6f the color scale for the photo counts should be the same for DMSO, PF10231066 and XL184, the best could be if Fig 6e has the same scale.

Response3:

Thank you for your helpful suggestion. We have adjusted the color scale of Figure 6f to be consistent with Figure 6e, the range is 500-10000.

Reviewer 2 Report

The Manuscript “Vascular niche facilitates acquired drug resistance to c-Met in-2 hibitor in originally sensitive osteosarcoma cells” by Tang et al describes the role of and anti c-Met drug, highlighting how the drug has limited effect on tumor growth, due to the overexpression of proliferation markers and angiogenesis proteins, such as VEGFR.

The manuscript provides a big amount of data. Nevetheless, I am somewhat sceptical about the overall general role of c-Met inhibitors combined with anti angiogenic drugs. The use of a drug (the anti c-Met) that leads to an increase in the Ki67 and VEGFR positivity generates some concerns in a tumor that has as a major caveat the generation of distant lung metastasis. The authors do not take into consideration a mouse model that generates metastasis, but it would be important to assess what happens to lung metastasis when the mice are treated with the anti c-Met drug.  

Despite interesting, the manuscript needs some revision.

There is something unclear about the western blots. The described antibodies detected total and phospho proteins of 145kDa. I have looked at the datasheet and, nor the reported antibodies, or other c-Met phospho antibodies report a phospho band of kDa65. Please refer to literature describing the 65kDa, or show the 145kDa phosphor-band described by the datasheet.

Please reorder the panels in Figure 2 so that the reader does not have to jump from one to the other.

Figure 2A: The authors state: “Immunofluorescence (Figure 2a) and western blotting analysis (Figure 2b) showed that the MNNG/HOS-OSCs presented elevated levels of c-Met phosphorylation compared with adherent MNNG/HOS cells.” Figure 2A does not correspond to the description. Figure 2A refers to a TUNEL assay (please clarify also the Figure Legend) and has nothing to do with c-Met phosphorylation. Moreover, I don’t understand the significance of the TUNEL assay shown. Images don’t show a difference in the presence of the green chanel when the CSC are treated with the drug. The CSC are smaller, but the relative green fluorescence is the same. Please quantifiy and show better images. Why do HOS monolayer cells are so green and apoptotic?

Figure 2C. Images are of very bad quality. Please provide better images.

Figure 2f. Again very bad quality. It is impossible to relate the graph in figure 2e to the images in figure 2f. Moreover, the graph in figure 2e is described as a crystal-violet assay, whereas the images are in brightfield. It would be clearer to have images corresponding to the same quantification of the graph.

Figure 2g. The TUNEL assay shown in figure 2g is completely different from the one shown in Figure 2a. How is the quantification of the TUNEL assay done? Do the authors normalize on the total number of nuclei? This is not described anywhere. Why is the percentage of apoptotic nuclei in the CSC higher than the 2D experiment (compare figure 1e and 2h). CSC are normally more resistant to drugs than 2D cells. Please justify.

Figure3. Please reorder the panels according to the description in the text.

Figure 3e. How has the phospho-Met signal been quantified? The magnification is so low that it is impossible to see any positive signal. Moreover the background white levels are very different between CTR and treated animals in figure 2c, which makes it even more difficult to understand whether there is a real difference in the phosphor-Met levels.

Figure 3d. The higher level o Ki67 seems problematic to me and I don’t fully believe that it might be due to proliferation of the tumor in the later stages of drug administration. If the authors want to state that The higher ki67 levels are due to the lack of effect of the drug in the later stages, the correct experiment to be performed is not the withdrawal experiment, but rather the opposite. They should keep the mice on the drug for a longer time (more than 20 days) and see whether the tumors rebound.

Figure 5a. The WB for c-Met shows a different result respect to the real Time. No downregulation of c-Met can be seen by WB.

Figure 5d. Please show also IF for totale c-Met.
